# Nuclear Receptor Coregulators in Hormone-Dependent Cancers

**DOI:** 10.3390/cancers14102402

**Published:** 2022-05-13

**Authors:** Hedieh Jafari, Shahid Hussain, Moray J. Campbell

**Affiliations:** 1Department of Molecular Genetics, The Ohio State University, Columbus, OH 43210, USA; jafari.15@buckeyemail.osu.edu; 2Department of Pharmaceutics and Pharmaceutical Chemistry, College of Pharmacy, The Ohio State University, Columbus, OH 43210, USA; sofi.2@osu.edu

**Keywords:** nuclear receptors, coactivators, corepressors, hormone dependent cancers, chromatin, transcriptome, epigenetic, prostate cancer, breast cancer, therapy resistance

## Abstract

**Simple Summary:**

Altered nuclear receptor signaling is well-established to contribute to a range of hormone-dependent cancers and therefore they are therapeutic targets, for example in breast and prostate cancer. However, nuclear receptor signaling in the cancer, and in response to therapeutic targeting is frequently altered in part by changes to the expression and function of the large numbers of coregulators that modulate sensitivity of the receptor, change their protein-protein interaction and alter the transcriptional consequences. These coregulators perform numerous roles in terms of regulating nuclear receptor responses and the functional diversity of these interactions is an area of emerging insight. Coregulator interactions, functions, and impacts on nuclear receptor signaling across hormone dependent cancers are the topic of this review.

**Abstract:**

Nuclear receptors (NRs) function collectively as a transcriptional signaling network that mediates gene regulatory actions to either maintain cellular homeostasis in response to hormonal, dietary and other environmental factors, or act as orphan receptors with no known ligand. NR complexes are large and interact with multiple protein partners, collectively termed coregulators. Coregulators are essential for regulating NR activity and can dictate whether a target gene is activated or repressed by a variety of mechanisms including the regulation of chromatin accessibility. Altered expression of coregulators contributes to a variety of hormone-dependent cancers including breast and prostate cancers. Therefore, understanding the mechanisms by which coregulators interact with and modulate the activity of NRs provides opportunities to develop better prognostic and diagnostic approaches, as well as novel therapeutic targets. This review aims to gather and summarize recent studies, techniques and bioinformatics methods used to identify distorted NR coregulator interactions that contribute as cancer drivers in hormone-dependent cancers.

## 1. Introduction

### 1.1. Nuclear Receptors and Hormone-Dependent Cancers

The healthy functioning of multicellular organisms requires highly coordinated gene regulation (both protein coding and non-coding RNA transcripts) to govern cell-fate decisions, sustain tissue organization, maintain homeostasis and respond to external stimuli, including towards dietary and xenobiotic compounds. These gene networks are directed by the combinatorial actions of *cis*-binding transcription factor (TF) complexes interacting with and impacting the underlying chromatin architecture. Transcriptional functioning of TF complexes is regulated further by *trans*-interacting events such as post-translational RNA and protein modifications and interactions with non-coding RNAs (ncRNAs). In this manner, TF complexes control gene expression by sensing and regulating chromatin architecture, and initiating RNA transcription, splicing, elongation and RNA modifications, as well as initiating feedback control of these very same processes. These regulatory steps result in precise and tunable protein expression in robust signaling networks that are central to homeostasis. Given these diverse functions, TF complexes are large, often being megadaltons in size, and the composition is highly dynamic. In addition, given the importance of TFs and their interacting proteins it is unsurprising that they comprise approximately 15% of the human protein-coding genome; ~1500 TFs, and ~1500 coregulators [1,2,3]. Clearly, all are not expressed and functioning in every cell.

To illustrate the impact of TF interactions with coregulators, it is informative to consider perhaps one of the best understood TF families, namely, the human nuclear hormone receptor (NR) superfamily, which consists of 48 members. The medical importance of NRs has been appreciated at least since rickets was first described in the 17th century by Daniel Whistler [4]; rickets arises from a lack of bone mineralization due to insufficient signaling via the NR, the vitamin D receptor (NR1I1/VDR) (reviewed in [5]). The expansive literature supports the concept that the NR network forms a major hormonal and environmental signaling conduit to the genome, the output of which is central to the development and homeostasis of many cells and tissues (reviewed in [6]).

An appreciation of the relationship between sex steroids and cancers of the reproductive system was pioneered by the work of Sir George Beatson in the 19th century, who began to define the relationship between estrogen and breast cancer (BRCA) risk [7]. Subsequently, in the 1940s this concept was echoed by the work of Dr. Charles Huggins and colleagues who established the endocrine synthesis of androgens and the relationship to prostate cancer (PRAD) [8]. Reflecting their biological importance and the appreciation of their functions, these TFs are major research targets in the arenas of hormone-dependent cancers, including not only BRCA (reviewed in [9]) and PRAD (reviewed in [10]), but also cancers of the ovaries (OVCA), endometrium (ECa), uterus (UCS), testis (TCGT), thyroid (THCA), pancreas (PAAD) and adrenal glands (ACC).

Outside of cancers, NRs impact other diseases including diabetes and metabolic syndrome; inflammation and inflammatory diseases; aging syndromes including bone health functions and cardiovascular diseases. Therefore, NR-targeting drugs include the well-established specific NR antagonists for BRCA and PRAD, and NR agonists for a wide range of inflammatory disorders and also to target specific leukemias. As a result, approximately 15% of prescription drugs target NRs; furthermore, NR-driven modulation of xenobiotic-metabolizing capacity is central to the clearance of an even bigger proportion of pharmacological drugs [11,12].

Across hormone-dependent cancers there is now a considerable volume of pre-clinical and clinical findings to define how NR-transcriptome mechanism functions are altered in cancer progression and central in response to therapeutic approaches. For example, prostate differentiation and homeostasis is intimately regulated by the androgen receptor (NR3C4/AR). Indeed, in 1972, AR control of cell-fate decisions was the catalyst for defining the field of programmed cell death [13]. Transcriptomic approaches, in the first case using microarrays and subsequently applying next generation sequencing (NGS) approaches, have established that PRAD cells display a remarkable diversity of AR transcriptional responses, and underscored that there are few examples of obligate and universal transcriptional responses regulated by the AR [14,15,16]. Widespread epigenetic events redirect AR genomic binding, including the recommissioning of embryonic enhancers [17]. As a result, in PRAD the AR no longer regulates luminal differentiation and alternative programs are enhanced, leading to novel lineages such as neuroendocrine (NE)-PRAD [16,18,19,20,21,22,23,24,25,26,27,28,29,30,31]. This finding is echoed by the diverse responses of estrogen receptors (NR3A1/ER⍺, NR3A2/ERβ) in BRCA. Many investigators have considered how NR coregulators are altered to explain how NR functions appear to be extremely context-dependent and are redirected through cancer progression [32,33,34,35,36,37,38,39].

### 1.2. A Primer on Nuclear Receptors and Their Interaction with Regulatory Complexes

The NR superfamily is structurally classified into seven families by sequence homology and, broadly, these receptors comprise three distinct functional groups with regard to their interaction with ligands (reviewed in [40,41]). Type I NRs, also called classical or steroid hormone receptors, include the ERs and AR and interact with high affinity ligands. Other receptors include the progesterone (NR3C3/PR), glucocorticoid (NR3C1/GR) and mineralocorticoid (NR3C2/MR) members. Type IIa NRs include binding non-steroidal ligands, as well as broad affinity ligands, and include all-*trans* retinoic acid (NR1B1/RARα, NR1B2/RARβ, NR1B3/RARγ), 9-*cis* retinoic acid (NR2B1/RXRα, NR2B2/RXRβ, NR2B3/RXRγ), peroxisome proliferator activated receptors (NR1C1/PPARα, NR1C2/PPARβ, NR1C3/PPARγ) and VDR. A third class of NRs referred to as Type IIb are orphan receptors with unidentified ligands.

To interact with the genome, Type I NRs form homodimers and Type IIb form heterodimers predominantly with RXRs, whereas Type IIb orphan receptors bind to the genome frequently as homodimers or monomers. The classical view of the cellular distribution for different classes of nuclear receptors is that Type I NRs are sequestered in the cytoplasm in the absence of ligands and not associated with *cis* regulatory elements (CREs) on target genes. In the presence of ligands, they shuttle into the nucleus to bind CREs and regulate gene expression. By contrast, the Type IIa NRs are nuclear-resident bound to CREs independent of ligands where they exert a repressive effect, and ligand activation generally redistributes them and leads to transactivation. Type IIb orphan NRs are also nuclear resident and can move across the genome in response to various stimuli [40,41]. However, with the development of chromatin immunoprecipitation coupled with NGS (ChIP-Seq) it has become apparent that these views are too rigid and genomic binding of all NRs is detectable in the absence of ligands, albeit to a lesser extent for Type I NRs.

Certainly, NR genomic localization sets the stage for a wide range of NR–protein interactions, many modulated by ligand and protein interactions, and allow NRs to participate in a wide variety of *cis*- and *trans*-regulatory mechanisms to positively and negatively regulate transcription [42,43,44,45]. NRs bind to CREs in a protein–DNA complex, whereas NRs can bind indirectly, in *trans*, to CREs and regulate transcription indirectly in a protein–protein–DNA interaction complex. Other than proteins, non-coding RNAs such as miRNA and long non-coding RNAs (lncRNAs) can alter NR-regulated gene expression by directly affecting the DNA environment or other RNA-binding proteins (RBPs) [46]. For example, the steroid receptor activator (SRA) was one of the first regulatory lncRNAs identified to coactivate a NR, namely the GR [47,48]. The SRA also coimmunoprecipitated with the coactivator NCOA1/SRC1, suggesting a scaffolding function for a wider range of NRs (reviewed in [47]).

Structurally, NRs contain a variable N-terminal domain (NTD), a DNA-binding domain (DBD), a hinge region, a conserved ligand-binding domain (LBD) and a variable C-terminal domain. The two most highly conserved domains amongst all NRs are the DNA-binding domain and the ligand-binding domain. The DBD contains two zinc fingers for genomic binding [49,50]. The LBD comprises a three-layered, antiparallel, α-helical sandwich and within this α-helical core, two additional layers of helices form the ligand-binding cavity. An additional helix for ligand-dependent transcriptional activation (AF2) resides at the C-terminal of the LBD and adopts different positions depending on the presence or absence of ligands [51,52,53,54,55,56,57,58]. This domain is mainly responsible for the interaction with transcriptional coregulators depending on conformational changes induced by different ligands [59,60], as ligand activation induces an allosteric change, thus inducing activation [6,61].

The concept that protein–protein interactions regulate TF functions was established in early studies of yeast by Ptashne and Gann, which identified roles for TF adapter proteins to bring about successful mRNA regulation. These proteins were not considered as constitutive members of RNA pol II, but indirectly associate with TFs and could positively modulate gene expression, and hence, gave rise to the term “activators” [62]. This discovery then led to the idea of protein interactions modulating TF functions.

The canonical view of NR function is that in the absence of ligands the receptor associates with corepressors. Structural analyses of corepressors, such as nuclear receptor corepressor 1 and 2 (NCOR1/NCOR2), revealed they contain an analogous extended helix with LXXH/IIXXXI/L residues forming a common motif called the CoRNR-box (corepressor-nuclear-receptor), which is required for interaction with RXR, TR and other NRs [63]. The CoRNR-box occupies the same hydrophobic pocket contacted by the NR box in the absence of ligands, due to displacement of the AF2 helix [64,65,66,67].

The presence of ligands induces structural LBD conformational changes, dissociates corepressor complexes and exposes protein surfaces for coactivator complexes to bind. These complexes contain chromatin modifier proteins, which further remodel chromatin and facilitate the interaction between the NR complex bound at enhancer/promoter regions of target genes to the Mediator complex [68] and the basal transcriptional machinery at the TSS. Coactivators such as NCOA1/SRC1 interact with NRs through helical motifs that contain an LXXLL-consensus peptide motif, also called the NR box.

Agonist binding reduces the affinity of nuclear receptors for CoRNR-box-containing corepressors and increases the affinity for LXXLL-containing coactivators. In vivo studies demonstrate a swing of agonistic or antagonistic activity for some BRCA treatments, such as tamoxifen. This effect is due to selective recruitment of coregulators to regulatory regions, depending on tissue specific expression level of either coactivator or corepressor [69].

Interactions between NRs and coregulators has been a topic of extensive investigation (reviewed in [70,71,72,73,74,75]). An intriguing question in the coregulator field is over how broad is the definition of their function. Clearly, coregulators that directly bind NRs are readily defined, but it is less clear if this extends to the cofactors that interact in a protein–protein manner with the NR complex, or function upstream of NR signaling. For example, the SWI/SNF complexes that remodel the nucleosome position may not directly inteact with NRs, but nonetheless regulate their function [76,77] and similarly downstream coregulators that facilitate the process of transcription, such as helicases are impactful [78,79]. In this manner, if the definitions of coregulators are extended, this results in a large number of proteins impacting NR coregulator proteins [80]. A working model of NR and coregulator functions is shown in Figure 1.

### 1.3. Functions Played by Nuclear Receptor Coregulators

Previously, we used bioinformatic approaches to develop [1] unique lists of coactivators, corepressors and mixed function coregulators that sum to more than 1500 proteins. Testing the enrichment of these components in NR complexes reveals the diversity of components. We have examined coregulator enrichment in NR-dependent RIME data [81], for example, to capture all VDR-interacting proteins, and identified at least 100 interacting proteins, many of which are classified as coregulators; thus, reflecting the findings of others [1,82,83,84,85]. This diversity of coregulator interactions, coupled with their dynamic sharing and sequestration, has been proposed to be a critical aspect of NR network behavior [86]. On the basis of function, coregulators can contribute to the control of the following events:

a. Nucleosome positioning. In humans the switch/sucrose non-fermenting (SWI/SNF) ATP-dependent chromatin remodeling complex, known as the BAF complex, includes coregulators that change nucleosome positioning through ATP-dependent helicases. These ATP helicases include various domains to bind chromatin; bromodomains (identify acetylated lysine), chromodomain helicase DNA-binding (CHD), plant homeodomain (PHD) domains (identify methylated lysine) and HAND-SANT-SLIDE domains (identify unmodified histone tails). Recently, another group of mixed function coregulators have been identified, which can act as a coactivator by associating with RNA pol II and corepressor by associating with nucleosome remodeling and the deacetylase (NuRD) complex [87,88].

b. Histone modifications. Histone modifying enzymes, such as acetyl transferases (e.g., KAT8) and histone deacetylases (HDACs) (e.g., HDAC1), alter the net charge of the specific residues on the histone tails within the nucleosome, thereby loosening or tightening the DNA–histone interactions. Activating modifications are typified by those on histone 3 (H3), such as acetylation of lysine 9 [89,90], which in turn results in gene transactivation. By contrast, depending on the site, histone methylation can both activate and repress transcriptional states of the chromatin. Coregulator components can recognize specific histone modifications, such as the SANT motif in the case of the NCOR2 complex [91]; thus, the genomic interactions of coregulators most likely reflect NR-targeted loci recruitment coupled with nucleosome affinity. Coactivator-associated arginine methyltransferase 1 (CARM-1), which is also known as protein arginine methyltransferase (PRMT-5), is a histone methyltransferase acting at H3R17, was identified initially by its ability to interact with NCOA coactivators. Due to its indirect recruitment to ligand-bound NRs, including the AR, CARM-1 has been classified as a secondary coactivator [92].

c. Post-translational protein modifications. More broadly, coregulators such as ubiquitin protein ligase E3A and CHY zinc finger domain-containing 1 demonstrate E3 ligase activity, which is implicated in the modulation of NR-driven regulation. E3A interacts with AR in a hormone-dependent manner and causes hormone-dependent recruitment to the promoter region of AR-target genes, leading to enhancing the transactivation function of the AR [93]. Similarly, SUMO homologs have been shown to affect AR-mediated transcription. SUMO-1 decreases, whereas SUMO-2 and -3 enhance AR transcriptional activity [94,95]. SUMO-3 interacts with the AR NTD and mutations in sumoylation sites of NTD are reported to increase AR transactivation. The positive effect of SUMO-3 on AR-transcriptional activity depends on stimulation by androgens and is independent of sumoylation sites of the AR or sumoylation by SUMO-3 [94].

d. RNA splicing and metabolism. The pre-mRNA splicing proteins polypyrimidine tract binding protein 1, paraspeckle protein 1 (PSP1) and PSP2 interact with AF-1 of the AR [96] and the non-POU domain-containing octamer-binding, another component of the U1snRNP pre-spliceosome, interacts with the AR NTD in a ligand-dependent manner and potentiates AF-1 function [96]. The metabolism of the cell can also impact NR function as powerfully illustrated by the significant literature on PPARs [97,98,99,100,101] and the PPAR coregulator PPARGC1a [1,102,103] in PRAD.

e. DNA repair. The protein kinase DNA-activated catalytic subunit (PRKDC) complex is involved in DNA repair and has emerged as a part of the transcriptional machinery. The AR-LBD interacts directly with the Ku70 and Ku80 regulatory subunits of PRKDC in a DNA-dependent manner. Ku70 and Ku80, as well as PRKDC, enhance AR activity in transactivation assays through the recycling of transcriptional factors [104]. 

f. Chaperones. Steroid hormone NRs bound to heat shock proteins (HSPs) are maintained in a stable and largely inactive configuration. For example, in the early stages of the AR activation process, AR-LBD interacts transiently with various HSPs, leading to an equilibrium in which the AR is maintained in an overall high-affinity ligand-binding state. Loss of the HSP70 cochaperone DNAJ HSP member A1 (DNAJA1) in a knockout (KO) mouse model leads to increases in AR protein levels and enhanced transcription of several androgen-responsive genes in Sertoli cells, and indeed, can be considered as a negative regulator of transactivation by the AR [105].

g. Signal integrators and transducers, scaffolds and adaptors. Several cellular scaffold proteins have coregulatory properties. For example, the endocytic coregulator protein huntingtin-associated protein 1 (HAP1) interacts with the AR-LBD [106]. In addition, caveolin-1 also interacts with the AR in the presence of a ligand [107]. Other proteins, such as the transforming growth factor beta 1 induced transcript 1 (TGFβ1I1), are involved in cell–cell contacts and also bind to AR-LBD and enhance transcriptional activity [108]. Indeed, protein tyrosine kinase 2 beta phosphorylates TGFβ1I1, which in turn represses AR transactivation [109]. The TGFβ1|1-related protein paxillin (PXN) also localizes within focal adhesions and can participate in AR signal transduction pathways. PXN directly interacts with the AR, and overexpression of PXN results in increased shuttling of the AR to the nuclear matrix [110]. Similarly, the TACC family paralogs are known to interact with histone acetyl transferases [111] and RAR complexes [112,113], and also participate in microtubule integrity [114]. Dysfunction of TACC1 is implicated in several cancers including BRCA [115,116,117], but is under-explored in PRAD, although it is associated with worse disease-free survival [118].

## 2. Examples of Nuclear Receptor Interactions with Coactivators

In Table 1 are examples of coactivator genes, their superfamily they belong to, NR interactions and the frequency of structural variation in several hormone dependent cancers.

### 2.1. SWI/SNF (BAF) Complex

Highly dense chromatin requires nucleosome remodeling to enable TF access. There are four families of nucleosome remodelers in eukaryotic cells, namely, SWI/SNF, ISWI, CHD and INO80 [138], and most eukaryotes contain at least one of these four. The SWI/SNF complexes are multi-subunit, ATP-dependent chromatin remodelers that mediate nucleosome dynamics. He et al. demonstrated that the catalytic subunit of SWI/SNF, SMARCA4, binds to the base of the nucleosomes and the ATPase modules [139]. Other core subunits are either the Brahma (BRM) or Brahma-related gene 1 (BRG1) that act as the ATPase core, bind to acetylated histones, have been known as tumor suppressors and potential binding partners for VDR [140]. More recently, we revealed that BAZ1A, a SMARCA5 binding partner, governs VDR gene regulation [83]. Many mutations and the functional disruption of the SWI/SNF complex have been reported from cancer genome sequencing in 25% of all cancer types [141] and mutations in SMARCA4 have been reported in BRCA, PRAD, OVCA and PAAD [142,143]. Both SMARCA4 and its paralog, SMARCA2, are identified as tumor suppressors or drivers of cancer and emerging therapeutic approaches are targeting these genes [144,145,146]. These SMARCA mutations appear to impact the expression of genes including NRAS, KRAS and TP53 [147].

Key questions are whether SWI/SNF subunits are recruited to specific loci through their direct or indirect interactions with NRs and if they are essential for all NR-mediated transcriptional activation [76,148]. For example, NR4A1/NUR77 interacts with SWI/SNF subunits, mediates an opposing role to PPARγ and promotes AR-dependent transcription [76,149]. In PRAD (LNCaP cells), the expression of AR-target genes requires the recruitment of SWI/SNF complex proteins to proximal and distal regions or AR CREs in the presence of DHT [150]. Moreover, studies on BRCA relapsed tumors highlight the importance of SWI/SNF-related genes, such as ARID1 and ARID2, in chemotherapy and survival outcome of patients with ERα positive BRCA [151,152]. Other studies have shown the importance of the SWI/SNF complex for ligand-dependent transcriptional activation of the glucocorticoid receptor (GR) using a hormone-responsive mouse mammary tumor virus (MMTV) promoter [153].

### 2.2. NCOA/p160/SRC

The earliest interaction discovered between NRs and regulatory proteins involved NCOA proteins [154], which belong to the P160/SRC family. The isolated cDNA of NCOA1/SRC1 was first identified through a yeast 2-hybrid assay for PGR [155] and was mapped to the 2p23 chromosome [156]. Similarly, in BRCA, frequent copy number gains were shown to encapsulate amplified in breast cancer 1 (AIB1), which was subsequently identified as NCOA3 [157,158].

Structurally, the NCOAs are composed of the basic helix–loop–helix-PER-ARNT-SIM (HLH-PAS) domain on N-terminus, a serine/threonine-rich (S/T) domain and a nuclear receptor interaction domain (NRID) in the middle and two transactivation domains (AD) on the C-terminus [159,160]. To function as coactivators, NCOA1-3 associates with histone acetyl transferase proteins such as p300/CBP, p/CAF and the Mediator complex [154].

Overexpression of NCOA1, NCOA2 and NCOA3 has been reported widely in hormone-dependent cancers, including in BRCA, PRAD and OVCA [161]. NCOA3 splice variants are also potent in BRCA. NCOA3/AIB1Δ4 with truncated exon 4 has recently been shown to drive aggressive BRCA, in part by altering enhancer usage [162]. RNAi-mediated depletion of NCOA3 revealed that it is important in the proliferation of hormone-dependent and -independent PRAD and BRCA cells [163,164,165,166,167]. Moreover, the opposing function of NCOA1 and NCOA2 were shown in KO animal studies, in which they differentially altered the PPARγ2 transcriptional capacity [168].

High expression of these coactivators has also been linked to relapse after hormone-deprivation therapies by activating the mitogen-activated protein (MAP) kinase pathway that, for example, results in both ERα and NCOA3 phosphorylation [166,169,170,171,172,173,174]. In turn, recent studies have suggested that NCOA1 phosphorylation initiates cascades that activate the STAT3 pathway leading to therapy-resistant BRCA [173,175,176,177]. Additionally, NCOA3 recruitment to ERα at specific enhancers/promoters is disrupted in MCF-7 cells that are resistant to tamoxifen [178,179].

Perhaps unsurprisingly, in PRAD cells, knockdown of NCOA1 impacts AR capacity to regulate target gene expression. Specifically, in androgen-responsive LNCaP cells and C4-2 cells, which are androgen deprivation therapy recurrent variants of LNCaP, depletion of NCOA1 results in reduced growth and lower expression of AR-regulated genes. On the other hand, AR-negative PRAD cell lines PC3 and DU145 display no impact of NCOA1 depletion, supporting a role for NCOA1 to control AR function [180]. Reflecting these cancer-driver effects, NCOA3 has emerged as an attractive target for novel cancer therapeutics [181].

### 2.3. Transforming Acidic Coiled-Coil Containing Protein 1 (TACC1)

TACC1 belongs to the TACC family and has two other paralogs in humans, TACC2 and TACC3. Genetically, all TACC family proteins have different splice variants; for example, the TACC1 gene has 10 different transcriptional splicing variants with isoforms showing tissue-specific differences. Initially, TACC1 and TACC2 were detected to be involved in the progression of solid tumors [182], whereasTACC3 KO in a mouse causes a lethal embryonic phenotype and increases apoptosis [183].

TACC1 is located on chromosome 8p, a common amplicon in BRCA [184] that also contains MYC, CCND1 and ERBB2 [116]. In metastatic BRCA, the binding of an RNA-binding protein, called muscle bind-like 1, to 3`UTR of TACC1 suppressed metastasis by increasing the stability of the mRNA [117]. TACC1 functions to stabilize microtubules, control mRNA maturation and chromatin remodeling for unliganded NRs (including TR and RARs) [185,186,187] and alter their target gene expression [113] such that depleting TACC1 causes a reduction of RARα and TR ligand-dependent transcriptional activity. This broad list of TACC functions reflects the large number of protein-binding domains and underscores their roles in gene expression regulation.

Early studies detected the oncogenic nature of the TACC family with specific splice variants that promote breast tumorigenesis in mouse models [188,189,190], and expression levels that differ in specific cancer types; for example, being frequently overexpressed in BRCA, but with both up- and down-regulated expression in PRAD [118,185]. In PRAD, reduced expression of RARγ leads to increased cell proliferation by unopposed AR signaling; thus, targeting RARγ is therapeutically attractive [191,192]. Our studies indicate miR-96 targets RARγ and TACC1 and associates with aggressive PRAD tumors [118].

### 2.4. PPARG Coactivator 1 Alpha (PPARGC1A)

PPARGC1A was first cloned in a mouse in 1998 and possesses a LxxLL motif and a serin- and arginine-rich domain with a PPARγ-binding site in between them and an RNA-binding site on its C-terminus [193]. This protein acts as coactivator for NRs such as PPARγ, AR, ERα, ERβ and TR, and controls a variety of mechanisms, including those involved with mitochondrial biogenesis [194]. The activity of PPARGC1A is regulated through post-translational modifications by acetyltransferase and deacetylase (GCN5 and SIRT1, respectively) with the acetylated form being inactive [195,196]. Generally, the expression level of PPARGC1A depends on the energy demand of the cell, and therefore, acts as master regulator of mitochondrial biogenesis. Other pathways such as adaptative thermogenesis, glucose and fatty acid metabolism and circadian rhythm are coordinated through the PPRGC1A coactivator [197].

Unsurprisingly, given the roles in energy regulation, PPARGC1A impacts tumor progression and growth, but the precise mechanisms are cancer-specific. PPARGC1A is overexpressed in BRCA, helping in survival, proliferation and stem cell maintenance, whereas in PRAD there are examples where the reduced expression of PPARGC1A is associated with aggressive features [1,103,198,199]. However, in other settings, such as PPARγ overexpression, cooperation with PPARGC1A increases AKT levels which in turn results in mitochondrial biogenesis that drives PRAD progression [97]. Furthermore, in the BRCA cell lines MCF7 and MDA-MB-231, overexpression of PPARGC1A induces apoptosis and inhibits glycolysis metabolism [200]. Additionally, PPARGC1A exerts its function through the regulation of estrogen-related receptor alpha (ERRα) activity and inhibits PRAD metastasis with metabolic reprogramming of the cells [103]. Interestingly, the genomic analysis on metastatic PRAD samples is in line with the idea of a selective pressure on PPARGC1A deletion as the disease progresses [201,202,203]. The conditional deletion of Pgc1α in a murine model’s prostate epithelium together with Pten, a tumor suppressor, increased the prostate mass and invasiveness of cancer. Rewiring the metabolic landscape of the PRAD cells is possible through the reprogramming of the signaling pathway and PPARGC has been reported to coordinate complexes, although the detailed mechanism has yet to be studied [103].

## 3. Examples of Nuclear Receptor Interactions with Corepressors

In Table 2 are examples of corepressor genes, their superfamily they belong to, NR interactions and the frequency of structural variation in several hormone dependent cancers.

### 3.1. RE1-Silencing Transcription Corepressor (REST) Complex 1 (RCOR1)

RCOR1 contains two SANT domains separated by 191 aa [226]. RCOR1 is part of a complex with HDAC1, -2, BHC80, high mobility group 20B (HMG20B) and lysine demethylase 1A (KDM1). KDM1A identifies and causes demethylation at histone 3 lysine 4 (H3K4), leading to gene repression, although demethylation of H3K27 leads to gene activation at H3K4 and K27 [227,228]. KDM1A’s assembly into the RCOR1 complex promotes demethylation of H3K4me2 at active loci and the subsequent recruitment of MECP2 to facilitate CpG island methylation and the generation of a repressive chromatin structure [228,229]. 

For example, this complex mediates AR actions on gene repression. Imaging has revealed REST and AR closely colocalize in vivo [206]. Another mechanism of REST-mediated transcriptional repression involves chromodomain on Y-like (CDYL), which acts as a REST corepressor that physically bridges REST and the histone methylase G9a to repress transcription [207]. RCOR1^−/−^ mice are profoundly anemic and die in late gestation. Mutant erythroid progenitor cells derived from knockout mice form myeloid colonies instead of erythroid colonies, indicating a change to the myeloid phenotype [230]. 

Forkhead box (FOX) A1 recruits the KDM1A/RCOR1 complex and HDAC-1/2 to the AR-regulated enhancers and suppresses basal transcription in an AR-independent manner [25482560]. AR and REST repressive complexes may also contribute to repressing the transcriptional regulation of the serine peptidase inhibitor, Kazal type 1 (SPINK1) [204], and indeed, the AR and REST function as transcriptional-repressors of SPINK1 and AR antagonists alleviate this repression leading to SPINK1 upregulation, which in turn is linked to transdifferentiation, stemness and cellular-plasticity [204]. 

### 3.2. NCOR1 and NCOR2 Corepressor Complex

NCOR1 and its paralog NCOR2/SMRT were discovered separately in 1995 as part of efforts to explain gene repression by NRs. NCOR1 and NCOR2 bind to TR and RARs to inhibit the expression of their target genes in the absence of ligands [231,232]. Ncor1^−/−^ mice with targeted deletion of the 5′ SIN3 interaction domain exhibit defects in the development of erythrocyte, thymocyte and neural events [233]. Tissue specific deletion of NCOR1 in muscles leads to enhanced exercise endurance due to an increase in muscle mass and number of mitochondria, indicating that loss enhances the functioning of muscles, as a result of its impact on PPARs signaling [234]. Ncor2^−/−^ mice have an impaired development of the forebrain, and the corepressor is required for maintenance of the neural stem cell state [235].

Both NCOR1 and NCOR2 have N-terminal repressor domains (RDs) that recruit and associate with additional proteins, forming a large corepressor holocomplex. For example, RDs interact and recruit SANT-like domains (Swi3, Ada2, Ncor1, and TFIIB) to synergistically promote deacetylation through a deacetylase activation domain and histone interaction domains on histone tails through C-terminal RIDs that tether the corepressor holocomplex to the NR partners on target genes [236,237,238,239]. HSPA8 and TCP1 chaperones occlude the active site for HDAC3. NCOR1 and NCOR2 corepressor complexes interact with HDAC3, 4, 5 and 7 through the deacetylase domain of the NCOR1–NCOR2 complex, leading to the expulsion of two chaperone proteins and freeing the active site of the HDAC [240]. Mechanisms have emerged whereby activated TFs can recruit NCOR1 and NCOR2, leading to active gene silencing. For example, ligand-dependent sumoylation of the PPARγ ligand-binding domain recruits the NCOR1 and HDAC3 complex to drive transrepression [241]. In turn, this prevents NCOR1 recruitment to the ubiquitylation/19S proteasome and promoter clearing.

These corepressors undergo significant alternative splicing to generate protein variants that determine function [242,243]. For example, NCOR splice variants RIP-13α and RIP-13𝛿1 directly interact with TF_II_B, TAF_II_32 and TAF_II_70 [244]. The most prominent alternative-splicing events occur at the receptor interaction domain (RID)-3 and alters the affinity for NR partners [242,245]. Furthermore, NCOR2 is regulated by mitogen-activated protein kinase kinase kinase 3 (MAP3K3) cascades that induce its release from its receptor partners, its export from nucleus to cytoplasm and its derepression of target gene expression. Cells can customize their transcriptional response to the MAPKKK cascade signaling by selective corepressor splice variant expression, and by selective exploitation of specific tiers of the MAPK cascade [246,247,248,249]. 

### 3.3. SIN3 Transcription Regulator Family Member A (SIN3A) 

The SIN3 gene was first cloned in *S. cerevisiae*. SIN3A has four paired amphipathic alpha-helix (PAH) motifs which are imperfect repeats of ~100 aa residues [250,251,252]. Based on a primary sequence analysis, each PAH motif was proposed to contain two α-helices; one exclusively hydrophilic, the other hydrophobic, similar to the MYC family of transcriptional factors within the helix–loop–helix protein dimerization domains even without possessing significant homology at the amino acid level, indicating Sin3’s potential for a multitude of protein–protein interactions [253]. SIN3A^+/−^ mice have normal expression of SIN3A, whereas SIN3A^−/−^ do not survive post-embryonic day 6.5, indicating the essential role of SIN3A in early development at the peri-implantation stage [254].

SIN3A is capable of multiple protein–protein interactions. SIN3 is devoid of intrinsic DNA-binding abilities and its ability to target chromatin comes through direct interactions with the DNA-binding TFs, such as Mad1 or KLF repressors, and indirect interactions with adaptor proteins including NCOR1/2 [255]. It is thought that the SIN3 proteins serve as a scaffold on which corepressor complexes assemble and recruit HDAC activity. The TGIF interacts with the DBD of AR, facilitating the binding of SIN3A and HDAC1 to form a repression complex and resulting in deacetylation of the core histones. Repression of AR-mediated transcription by TGIF indicates a potentially significant mechanism to control androgen-induced cell growth, which may play a critical role in the growth and development of normal prostate tissue, and in the development and progression of PRAD [216]. RNA splicing factors, splicing factor proline- and glutamine-rich (SFPQ) and non-POU domain-containing octamer-binding (NONO) interact and attract SIN3A synergistically to form protein complexes with the AR in a ligand-independent manner and inhibit its transcriptional activity [256].

The CCCTC-binding factor (CTCF) comprises autonomous silencing domains that mediate transcriptional repression when tethered to a promoter sequence [257,258]. At least one of these domains, the zinc-finger region of CTCF, binds to SIN3A without binding to NCOR1/NCOR2 and recruits HDAC [257]. Two SIN3A domains interact independently with the CTCF zinc-finger cluster. The ability of CTCF regions to retain deacetylase activity is correlated with the ability to bind to SIN3A and to repress transcription. Taking these results together, the synergy in repression mediated by the NR, TR and CTCF is achieved by the binding of multiple molecules of SIN3A to the TR/CTCF–DNA complex, thus providing a large platform for the recruitment of histone deacetylases [259]. 

SIN3A also acts as a regulator of gene expression, survival, and growth in Erα-positive BRCA cells [260]. For example, reduction of SIN3A expression by RNA interference specifically inhibits estrogen-induced repression of ERα [261]. Furthermore, the SIN3A corepressor complex interacts with a long non-coding RNA NEAT1 to physically associate with FOXN3 in ER+ BRCA cells. The FOXN3–NEAT1–SIN3A complex represses genes that are critically involved in EMT, including GATA3. The SIN3A–FOXN3–NEAT1 complex transrepresses ERα itself, forming a negative-feedback loop in transcription regulation [214]. The SIN3A/ZNF704 corepressor complex also regulates the circadian clock and, in this manner, is a further potential driver for breast carcinogenesis. ZNF704 is a transcriptional repressor that mediates its actions by associating with the SIN3A complex. The ZNF704/SIN3A complex represses a panel of genes, including PER2, that are critically involved in the function of the circadian clock. ZNF704 promotes the proliferation and invasion of BRCA cells in vitro and accelerates the growth and metastasis of BRCA in vivo [212]. 

SIN3A is able to interact with other factors outside its PAH domains. For example COPS2/ALIEN-mediates silencing through the recruitment of SIN3 to its N-terminus and interacting with DNA-bound transcriptional silencers at the C-terminus. This functional interaction has been demonstrated with VDR on the human CYP24 promoter containing VDRE [262]. COPS2 also interacts in a hormone-sensitive manner with TR to function as a corepressor [263].

### 3.4. Ligand-Dependent Nuclear Receptor Corepressor (LCOR)

The LCOR is a transcriptional corepressor, first cloned from a fetal brain, with high expression in the liver, testis, ovary, kidney and brain. The LCOR is recruited to agonist bound-NRs through a single NR box. The LCOR comprises a NR motif, a nuclear localization signal, tandem PXDLS motifs and a HTH domain [264]. 

The LCOR binds to the LBDs on the ERα/β, AR, GR, RAR α/β/γ, RXRα, PPARγ, VDR and PGR in a ligand-dependent manner and acts to attenuate signaling by agonist-bound receptors [220,265,266,267,268]. LCOR-mediated repression of ERα and GR is inhibited by use of the HDAC inhibitor trichostatin A (TSA), but repression of VDR and PR does not happen in the presence of TSA, which indicates that the LCOR functions as a corepressor in an HDAC-dependent and independent manner. LCOR N-terminal PXDLS motifs recruit CTBP 1 and 2 corepressors, which are responsible for their corepressor functioning [221]. LCOR and CTBP1 colocalize in nuclear bodies which also contain the CTBP-interacting protein (CTIP) and polycomb group repressor complex marker BMI1. LCOR function is strongly dependent on the HTH domain, and its deletion completely abolishes the LCOR corepressor function. The LCOR, CTBP and CTIP are recruited endogenous PR- and Erα-stimulated genes in a hormone-dependent manner. SiRNA-mediated knockdown of the LCOR/CTBP1 helped in the expression of PR- and Erα-stimulated reporter genes as well as in endogenous PR-stimulated target genes, whereas complete deletion resulted in gene-specific effects on Erα-regulated transcription, which mostly was reduced gene expression. Thus, the LCOR and CTBP1 usually act as repressors of PR-related transcription, but in some cases can enhance the transcription of some genes [266]. 

In BRCA, the LCOR has two mRNA variants called LCOR and LCOR2. Both LCOR and LCOR2 inhibit estrogen-mediated signaling and BRCA cell proliferation via the HTH domain. Both the LCORs bind with NRIP1 via the LCOR HTH domain and C-terminal domain (CTD) of NRIP1 and this interaction is crucial for inhibition of gene expression and cell proliferation [269]. 

The LCOR acts as a corepressor for the AR and inhibits PRAD in vivo. The LCOR interacts with the AR and is recruited to chromatin in the presence of AR. However, the NR box of LCOR is dispensable for interaction with the AR and LCOR and instead interacts with the DBD of the AR on chromatin. In androgen-independent PRAD, the interaction between the LCOR and AR is inhibited by signaling pathways involving SRC [220]. The CTD of the LCOR interacts with KLF6 in multiple cancers and they bind together to the promoters of genes, including CDKN1A. The LCOR contributes to repression mediated by KLF6 in a promoter- and cell type-dependent manner. SiRNA-mediated knockdown of LCOR, CTBP1 and KLF6 in PRAD cell line PC3 induced expression of CDKN1A, CHD1 and other KLF6 target genes [270]. 

## 4. Cointegrators

To complement these roles for coactivators and corepressors, it has become clear that these protein complexes link to several co-called cointegrator complexes. For example, the CREB-binding protein (CBP) was initially identified as a coactivator of multiple transcriptional activators, including p53, NF-𝜅B and NRs [271]. The CBP forms a ternary complex with NCOA family members and NRs; for example, the CBP synergizes with NCOA1 in the transactivation of ERα and PR [272]. Direct interactions between the CBP and RXRs and TR occur via the N-terminal of CBP, which comprises the NR box required for receptor interaction [273,274]. The interaction between NCOA1 and unliganded ER and PR are relatively weak in comparison to the recruitment of other NCOA1 complex proteins to ER and PR [275,276]. Overall, it indicates that the CBP acts as a common limiting cofactor for diverse TF/coactivator complexes, acting as a cellular cointegrator to collate multiple different signals into an integrated response at promoters containing multiple *cis*-acting elements [274]. Similarly, activating signal cointegrator-2 (ASC-2) has an indirect binding site for the AR, in addition to two LXXLL motifs which interact with other NRs. ASC2 stimulates AR transactivation, whereas the retinoblastoma protein hinders transcriptional activation of the AR. ASC2 thereby has three distinct nuclear receptor interaction domains [277].

## 5. Disrupted Coregulators Function in Therapy-Resistant Hormone-Dependent Cancers

A number of different selective antiestrogens (e.g., tamoxifen) and aromatase inhibitors (e.g., anastrozole) have been developed for hormone-sensitive ER+ BRCA. Drugs with a combined activity of aromatase inhibition and ER degradation activity, such as fulvestrant, have also been approved against ER+ and HER2- advanced BRCA. Resistance to these therapies in BRCA is attributed to the retention of ER signaling, which is mediated by the interactions between activated ERα and coregulator proteins [278,279,280,281].

A role for EZH2 has emerged, which mediates epigenetic silencing of the ERα cofactor GREB1 and which causes ERα coregulators to reallocate on chromatin to drive the conversion of antiestrogen into an agonist [282]. Furthermore, adaptive mutations in the ligand-binding domain of ERα in response to endocrine therapy render them with high transcriptional activity even in the absence of estrogen [283,284]. This happens mostly due to the ability of the mutant ER proteins to interact with coregulator proteins to further contribute to endocrine therapy resistance [285]. Mutations in ERα are also associated with estrogen insensitivity as they affect the interaction between ligands and the recruitment of coactivators, such as NCOA1, NCOA2 and NCOA3, to drive metastatic BRCA [286]. A mutant ERα recruits coactivators in the absence of estrogen to confer antiestrogen resistance by giving rise to an altered antagonist state that resists inhibition within the ERα helix 11 and helix 12 [287].

More widely, other coactivators involved in tamoxifen resistance include TIMELESS, which binds to ERα to enhance its transcription and when overexpressed it increases PARP1 expression and ER-induced gene regulation through its proximal LXXLL motifs [288].

Strong parallels exist in the emergence of ADT-R-PCa in response to drugs, including enzalutamide, fulvestrant, flutamide and darolutamide. In half of the PRAD patients, AR gene amplification has been detected among the most frequent genomic changes for recurrent cancer [203,289]. NCOA2/SRC-2 mRNA levels are elevated in 20% of primary and 63% of metastatic PCA, which are associated with worse recurrent-free survival, whereas sensitivity to bicalutamide can be restored by the genetic KO of SRC-2 [290]. Various coregulators are implicated in changing cells’ response to therapy. The AR coregulator, the serum response factor (SNF) inhibitor, can modulate the cell response to enzalutamide in hormone-resistant cells [291].

By contrast, 5–10% of recurrent PRADs have loss-of-function mutations in corepressors, such as NCOR1 or NCOR2, that enhance the enzalutamide resistance. Altered expression of NCOR1 and NCOR2 has been reported across several different hormone-dependent cancers including BRCA, PRAD and TRC [32,33,292,293]. It is unclear what drives NR-selectivity, with NCOR1 regulating PPARγ and elevated NCOR1 expression, and activity-distorted PPARα/γ gene targets involved in cell cycle control [33]. NCOR2 has some selectivity for the VDR, as NCOR2 mutants defective in the VDR interaction domain are unable to repress endogenous VDR-target genes [294]. Elevated NCOR2 levels in PRAD cells also disrupt VDR signaling, resulting in suppression of target genes associated with antiproliferative action and 1α,25(OH)_2_D_3_ sensitivity [295]. By contrast, elevated NCOR1 mRNA in BRCA cells correlates with the suppressed regulation of VDR-target genes and the ability of cells to undergo arrest in the G(1) phase of the cell cycle. A similar increased ratio of corepressor mRNA to VDR occurred in matched primary tumors and normal cells, noticeably in ERα negative tumors. Combinations of vitamin D3 compounds with TSA restored VDR antiproliferative signaling to levels which were indistinguishable from MCF-12A cells [32].

A reduction in NCOR1 levels causes bicalutamide resistance in LNCaP cells and murine PRAD in vivo [38]. PRAD cell lines treated with HDAC inhibitor sodium butyrate (NaB) form the AR-NCOR2/NCOR1 complex binding on the PSA promoter, indicating that AR and NCOR1/2 corepressors may form a stable complex in vitro and NaB may facilitate the interaction between the AR and NCOR2 [296]. In PRAD cell models, loss of NCOR2 leads to methylation at sites distal to transcription start sites and CpG islands, which phenotypically contributes to a neuroendocrine phenotype [63]. More recently, it has been reported that the reduced expression of NCOR2 increases PRAD progression and shorter disease-free survival and accelerates disease progression following androgen deprivation, suggesting a more direct role to regulate the AR [63].

## 6. Technology and Computational Approaches to Discover Coregulator and Nuclear Interactions

The central role of coregulators and TF interactions is to fine-tune and integrate endogenous and exogenous signals to impact cell-fate decisions. The regulatory interactions are highly specific and depend on the structural and physico-chemical characteristics of protein domains. There are a myriad of coregulator-dependent mechanisms of altered gene expression patterns in hormone-dependent cancers, and it is challenging to define cancer-driver mechanisms with these alterations. Germline and somatic structural variations alter TFs and coregulators, and the regulatory regions they bind [297,298]. In parallel, alterations to the epigenome [30] and 3-D genome [299,300,301] change the accessibility and responsiveness of gene promoter/enhancer regions. However, altered TF-enhancer relationships have potential to be targeted through selective epigenetic therapies [302,303,304], or the downstream gene networks may be uniquely drug-sensitive [305,306]. Furthermore, disease-specific enhancers provide the rationale for targeted tumor deep sequencing to dissect the interactions of germline and structural variation [307,308]. Therefore, there is considerable interest in identifying and dissecting TF-coregulator functions in hormone-dependent cancers.

### 6.1. Two-Hybrid Assay

One of the most utilized techniques that facilitated the discovery of protein–protein interactions is a two-hybrid assay that resulted in the discovery of NCOR1, NCOR2, NCOA1/SRC1 and many more coregulators. This technique requires using a reporter gene for expression control and fusing both the bait and the prey proteins with the DNA-binding domain (DBD, also called bait) and the transcriptional activation domain (AD, also known as prey), respectively. The two-hybrid assay is one of the basic low-throughput experiments developed in yeast, but now it can be utilized in mammalian cells as well [309,310], leading to the discovery of many VDR, RARα and AR coregulators [294,311]. Recently, Yachie N et al. have reported a two-hybrid assay capable of screening a full matrix of proteins in a single multiplexed strain pool. The advanced high-throughput screens have recently been developed to study the entire genome. Two of the categories are matrix-based and library-based techniques.

In the matrix-based, different prey protein-expressing clones will be plated in different wells, and afterwards they will be mated with bait strains in order to form chimeric diploids. Depending on the reporter gene expression, the chimeric colonies will be selected. For example, an RARα interaction with NCOR1 is one of the many interactions resulting from a high-throughput matrix-based approach [312,313]. In the library-based approach, a comparison between each bait and a pool of random prey (cDNA fragments) is made and the resulting interactions will be evaluated with DNA sequencing.

### 6.2. Affinity Purification Followed by Mass Spectrometry (MS)

The tandem affinity purification (TAP) method uses the principle of tagging a target protein with a TAP tag, and then this fused protein gets expressed in yeast to form the typical complexes in the cells. After two steps of IgG matrix-protein bond cleaving and calcium calmodulin-coated beads incubation, the fragments can be analyzed using MS. This technique has the advantage of a higher-order interaction direction and is not limited to binary information. Some of the insight into NCORs involved in AR regulation is due to this technique [314]. Recently the proximity-labeling MS (APEX-MS) has developed as a complementary approach for the affinity purification assay. APEX-MS is an in-cell labeling method that, upon treatment with hydrogen peroxide biotin-phenol, gets converted into radicals that can form covalent labeling of the proteins in a 20 nm radius [315].

### 6.3. Rapid Immunoprecipitation Mass Spectrometry of Endogenous Proteins (RIME)

Given that coregulators modulate the NR’s transcriptional activity, scientists have developed a total protein interactome method to understand the role of NRs’ dysregulation in different diseases. RIME allows the study of complexes in their native interactome by formaldehyde fixation and immunoprecipitation of the target protein. In the last step, on-bead digestion reduces the contamination, and the resulting peptides are tested with MS to identify direct protein interactions to the target protein. The advantage of this technique is that there is no need for an extra tag to purify or overexpress the target protein, because RIME utilizes the endogenous proteins and mimics the biologically and clinically relevant interactomes. Complex interactions can be further investigated using the RIME data to model protein networks and the interaction of complexes with each other [316].

Other CRISPR-based approaches, such as CAPTURE [317] or C-BURST [318], have also been developed to measure the protein complex and chromatin interactions in a specific genomic loci. To date, these have not been used to study NRs as they may lack sensitivity and have been most optimally applied to regions such as telomeres or globulin locus where the protein heterogeneity is less and the protein concentration is higher. However, it is likely that these approaches will be refined to identify the specific NR complex components at specific loci.

### 6.4. Protein–Protein Monitoring Methods

These approaches usually result in low-throughput data collection but can result in more in-depth understanding of two specific protein interactions. Some of these techniques are: fluorescence resonance energy transfer (FRET), X-ray crystallography and NMR spectrometry [319]. FRET involves detecting two fluorophore-tagged proteins in close proximity to each other [320]. However, to skip the spectroscopic labeling of the target protein, surface plasmon resonance (SPR) has been developed to quantitatively identify ligand–receptor interactions while the receptor is immobilized [321].

All aforementioned techniques are examples of protein–protein interaction detection methods and scientists are still developing more advanced techniques. Each method has its advantages and disadvantages, but all of them can open entirely new avenues of research and novel ideas to better understand the protein functions.

### 6.5. Bioinformatics Approaches to Identifying Coregulators

The first goal of analyses of coregulators from a genome-wide perspective is to establish comprehensive lists of TFs and co-regulator genes. Previously, we applied a text-mining approach to capture gene sets in Gene Ontology (GO) terms that contained phrases including “positive control of transcription”, “negative control of transcription”, “co-activator” and “co-repressor”, as well as other curated collections; for example, Panther, which stands for Protein ANalysis THrough Evolutionary Relationships, and is a classification system that was designed to classify proteins (and their genes) in order to facilitate high-throughput analysis [322], and FANTOM, which was established to undertake functional annotations of the full-length cDNAs that were collected during the Mouse Encyclopedia Project at RIKEN [2]. From these GO terms, the HGNC gene name and ENSEMBL transcript IDs can be retrieved using bioinformatics approaches [323] and combined to gene lists cross-referenced for uniqueness of, for example, coactivators.

Bootstrapping permutation approaches can then be used to test if the proportions of each gene group are altered more than predicted by chance; for example, in TCGA data. For mRNA or protein expression, copy number variation or mutation status, Z-scores of normalized data can be calculated and tested whether the observed alteration frequencies within gene classes (e.g., coactivators) are altered significantly, or a vector of changes for all genes was calculated and the observed values (e.g., mean Z-score of expression) of a given class tested in comparison to all genes detected in each cancer cohort. Random sampling methods such as bootstrapping can be used to simulate the distribution of changes across the genome for statistical comparison [324]. Empirical *p*-values are then calculated based on the group position relative to the sampling distribution of the genome. Finally, within each group the most altered genes within a class can be focused on for multivariate modeling [1]. Similarly, text-mining approaches to PubMed searches would be able to test how the distribution of published coregulator studies reflects the incidence of their alterations in a given cancer, for example. Our experience to date suggests that the experimental investigation is not evenly distributed across the coregulators that are actually most impactful [1].

Such lists of coregulators can be used to annotate other high-dimensional datasets to investigate enrichment. For example, recently we undertook RIME analyses of NCOR2 [63] to identify interacting proteins, and then by cross-referencing we were able to define which of these proteins were a part of the so-called “long-tail” of oncogenic drivers in PRAD [325]. More broadly, it is an intriguing challenge to identify how cancer phenotypes could emerge from the interactions of multiple relatively rare mutants of coregulators either with strong oncogenic drivers, or by converging on shared complexes, as seen with SWI/SNF complexes [326]. This would reflect the concept of “BRCAness” in which multiple genomic alterations phenotypically converge [327]. For example, we focused on the mutation status in tumors in advanced PRAD [289] and were able to identify coregulators that bind to NCOR2, exert known epigenomic functions, and are mutated in advanced PRAD.

Similarly, the annotation of ChIP-Seq data for either a given NR or a coregulator with a ChromHMM algorithm [328] that defined underlying epigenetic states allows testing of the unique and overlapping genomic distributions in different epigenetic states (e.g., in active enhancers or at poised enhancers), in cells/treatments or in the presence of ROSE algorithm-defined super-enhancers [329,330]. Indeed, testing the overlap of target ChIP-Seq data with comprehensive datasets as contained in Cistrome DB [331] allows co-enrichment testing of hundreds of TF and histone modification ChIP-Seq datasets to reveal the extent of the enrichment with other nuclear receptors or coregulators. RNA-Seq undertaken in parallel treatments can be matched with cistromic data to define cistrome-transcriptome relationships and test their phenotypic associations, for example, by using Kolmogorov–Smirnov tests to examine differences in cumulative distribution plots for cistrome binding sites with respect to the nearest gene, and again using bootstrapping approaches to measure how the specific cistrome relationships associate with gene expression patterns [332].

Finally, to test how coregulators impact correlations between, for example, AR and AR-target genes, partial correlation analyses can be used [33] to test how the expression of coregulators impacts the correlation of AR to DHT-regulated genes and identify how coregulators selectively promote the expression of genes associated with advanced PRAD emergence.

## Figures and Tables

**Figure 1 cancers-14-02402-f001:**
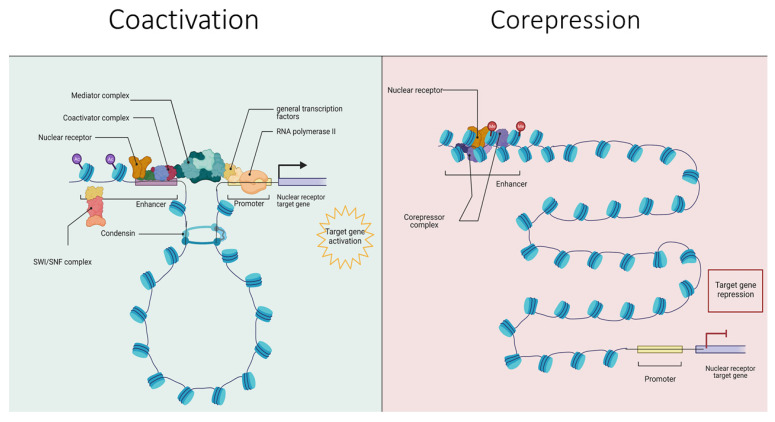
Schematic model of NR-target gene transcriptional activation (**left panel**) and repression (**right panel**).

**Table 1 cancers-14-02402-t001:** Examples of coactivators that interact with nuclear receptors and impact their functions.

Gene	Superfamily *	Functional Complex *	Example of NR Interaction **	Gene Defects ***	References
*SMARCA4*	PBAF complex	Nucleosome positioning	VDR, AR, PPARγ	BRCA—2.6% point mutations, 0.6% CNVs. CC—4.9% point mutations, 1.67% CNVs. OVCA—5.8% point mutations, 3.3% CNVs. PRAD—11.66% point mutations.	[119,120,121]
*TRRAP*	SAGA coactivator complex	Histone modifications	LXR, FXR, ERα,	BRCA—5% point mutations, 0.3% CNVs. CC—8.2% point mutations, 0.33% CNVs. OVCA—4.9% point mutations, 0.7% CNVs. PRAD—4.9% point mutations, 0.5% CNVs.	[122,123,124]
*CARM1*	7BS protein arginine methyltransferases	Histone modifications	ERα, AR, FXR, PPARγ,	BRCA—1.6% point mutations, 0.7% CNVs. OVCA—1.3% point mutations, 3.3% CNVs.	[125,126,127,128]
*NCOA1*	Lysine acetyl transferase	Histone modifications	RARβ, RXRα, PPARγ, ERα, TRα, GR	BRCA—6.1% point mutations. CAN—0.1% point mutations. PRAD—6.3% point mutations. OC—5.7% point mutations, 0.4% CNVs.	[129,130,131]
*PARP1*	Poly (ADP-ribose) polymerases	DNA repair, link to Mediator	PPARγ, ERα, NR4A4, RARβ, TR	BRCA—1.9% point mutations, 7.3% CNVs. CC—1.8% point mutations, 1.6% CNVs. OVCA—1.2% point mutations, 1.1% CNVs. PRAD—1.2% point mutations, 0.1% CNVs.	[132,133,134,135]
*TACC1*	TACC family	Signal integrators and transducers, scaffolds, and adaptors	THRβ, THRα, PPARγ, RARα, RARγ and RXRα	BRCA—3.7% point mutations, 8.8% CNVs. CC—1.2% point mutations, 1% CNVs. OVCA—2.8% point mutations, 1.7% CNVs. PRAD—2.8% point mutations, 0.7% CNVs.	[113,116]
*PPARGC1A*	RNA-binding motif containing	Coactivator complexes for metabolic programming	Interacts with PPARγ, PPARα, TR	BRCA—3.2% point mutations, 0.1% CNVs. OVCA—4.7% point mutations, 0.3% CNVs. PRAD—4.5% point mutations.	[136,137]

* All the information obtained from HGNC and UniprotKB websites, and literature review. ** Nuclear receptor-related interactions among top 25 interactions on STRING tool. *** Mutation percentages are obtained from COSMIC repository.

**Table 2 cancers-14-02402-t002:** Examples of corepressors that interact with nuclear receptors and impact their functions.

Gene	Superfamily *	Functional Complex *	Example of NR Interaction **	Genetic Defects ***	References
*RCOR1*	Myb/SANT domain-containing	Histone modifications	GR, AR	BRCA—2.9% point mutations, 0.1% CNVs. PRAD—3.0% point mutations. THCA—0.6% point mutations. OVCA—3.3% point mutations, 0.5% CNVs.	[204,205,206,207,208]
*MECP2*	Methyl-CpG binding domain-containing	Histone modifications	NCOR2, NCOR1	BRCA—2.3% point mutations, 0.5% CNVs. PRAD—1.4% point mutations, 0.2% point mutations. THCA—0.8% point mutations, 0.4 CNVs.	[209,210,211]
*SIN3A*	SIN3 histone deacetylase complex subunits	Nucleosome positioning	AR, ERα, RARβ, NR0B1/SHP	BRCA—2.7% point mutations, 0.1 CNVs. PRAD—2.4% point mutations. THCA—0.8% point mutations.	[212,213,214,215,216,217,218,219]
*NCOR1*	NCoR/SMRT transcriptional repression complex subunits	Histone modifications	AR, TR, RAR, VDR, PPARα/γ	BRCA—5.6% point mutations, 0.8% CNVs. PRAD—4.9% point mutations, 0.1% CNVs. THCA—1.8% point mutations. OVCA—3.4% point mutations, 0.1% CNVs.	[33,35,38]
*LCOR*		Histone modifications	ERα, ERβ, AR, PGR, RARα, RARβ, RARγ, RXRα VDR	BRCA—2.7% point mutations. PRAD—2.9% point mutations, 0.1% CNVs. THCA—0.3% point mutations. OVCA—2.7% point mutations, 0.1% CNVs.	[220,221]
*SFPQ*	RNA-binding motif containing	RNA splicing and metabolism	SIN3A, THRα, RXRα	BRCA—0.8% point mutations, 0.1% CNVs. PRAD—0.7% point mutations. THCA—2.1% point mutations.	[222,223,224,225]

* All the information obtained from HGNC, UniprotKB website and the literature review. ** Nuclear receptor-related among top 25 interactions on STRING tool. *** Mutation percentages are obtained from COSMIC repository.

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
