# Peer review of "Nuclear Receptor Coregulators in Hormone-Dependent Cancers"

_cancers, 2022, doi:10.3390/cancers14102402_

Round 1

Reviewer 1 Report

Authors have addressed the critical points in a full satisfactory manner.

Reviewer 2 Report

Manuscript is now in better shape and well improvised to be accepted.

This manuscript is a resubmission of an earlier submission. The following is a list of the peer review reports and author responses from that submission.

Round 1

Reviewer 1 Report

The review by Jafari et al., focusses on coregulators for the nuclear hormone receptor superfamily. In general, that is a field and not easy to review also considering the large number of coregulators known. Authors touch topics of coregulation of nuclear hormone receptors at the level of chromatin, posttranscriptional modifications, RNA, DNA repair, chaperones and signal transducers. Authors present two tables showing only selected examples of coactivators and corepressors.

Major comments:

  1. Considering that coregulation includes all the above-mentioned levels many factors are missing in this review. Such as IGF, EGF, PTEN-SRC-AKT, or KMTs and other factors are missing that can coactivate nuclear hormone receptors by phosphorylation or acetylation leading to enhanced nuclear translocation.

  1. In case authors indicate that coregulators are playing an important role at all these levels, many additional factors must be mentioned. Alternatively, authors may focus on transcriptional coregulators at chromatin only. Such a focused topic would be more concise.

  1. Authors mainly describe cell culture-based experiments. Important would be to include mouse knockout phenotypes whether a hint is provided for coregulation for nuclear hormone receptors, similar to that indicated for HSP40.

  1. What is missing is the role of coregulators dependent on the type of ligands that should include receptor antagonists. This is of course of special interest in hormone-dependent cancers (see title).

  1. For some lncRNA coactivator function was shown, such as SRC3, GAS5 and others. This is missing.

  1. The rationale showing in both tables only a very small selection of corepressors and coactivators is not clear. Why were these selected?

Should the list be expanded?

  1. Why the nomenclature of are some nuclear hormone receptors is provided as NRs e.g. NR3C1 (GR) while others eg. VDR?

Please present homogenously names of nuclear receptors.

  1. Some discrepancies exist between text and tables. E.g. LCoR is mentioned in text to be a corepressor of AR but not mentioned in the table.

Author Response

Reviewer #1:

General Comments:

We thank the reviewer for the thoughtful comments and the number of significant and interesting points raised and highlighting the difficulty of summarizing the pool of coregulators considering their wide spectrum of functions.

Major comments:

  1. Considering that coregulation includes all the above-mentioned levels many factors are missing in this review. Such as IGF, EGF, PTEN-SRC-AKT, or KMTs and other factors are missing that can coactivate nuclear hormone receptors by phosphorylation or acetylation leading to enhanced nuclear translocation.

Answer: Thank you for this insightful comment, and it was an issue that we debated greatly in the preparation of the review. While we agree that different post-translational modifications such as acetylation and phosphorylation can have effects on coregulator shuttling into the nucleus we have discussed SUMOylation as a post-translational modification, and the role it plays in coregulation (page, 5, top paragraph)

  1. In case authors indicate that coregulators are playing an important role at all these levels, many additional factors must be mentioned. Alternatively, authors may focus on transcriptional coregulators at chromatin only. Such a focused topic would be more concise.

Answer: We have aimed the major focus to be on coregulators that have their primary effect on chromatin but also believe the concept of non-chromatin-based functioning of protein coregulators to be intriguing and one that is worth highlighting to the reader. Therefore, we aimed to include the non-chromatin-based interactions to illustrate how those contribute to regulate transcription, as we believe these are areas that are emerging strongly in the literature.

However, to make this clearer we have clarified this (p.4, p.5 in red) and then focused and clarified both tables to illustrate how proteins from many of these steps are altered (Table 1 and Table 2).

  1. Authors mainly describe cell culture-based experiments. Important would be to include mouse knockout phenotypes whether a hint is provided for coregulation for nuclear hormone receptors, similar to that indicated for HSP40.

Answer: Thank you for the suggestion. We have surveyed different mice knockouts that have been reported for different coregulators, and reviewed and included some of the more pronounced the phenotypes of mice produced from such knockout experiments. (throughout the text, e.g. p.9, 2nd to last paragraph, p.12 first and third paragraph)

  1. What is missing is the role of coregulators dependent on the type of ligands that should include receptor antagonists. This is of course of special interest in hormone-dependent cancers (see title).

Answer: We thank the reviewer for pointing out the importance of nuclear receptor antagonists in the coregulator functions. In P.4 second paragraph, p.15 second paragraph this process is discussed at the structural level and we also added some information to draw the relationship between coregulators, their expression level and how they can affect current therapeutic approaches.

We have also included a new section summarizing examples of how coregulators change the response to hormonal therapies in these cancer (Section 5)

  1. For some lncRNA coactivator function was shown, such as SRC3, GAS5 and others. This is missing.

Answer: Thank you for your insightful comment. We have included some information regarding non-protein regulators of nuclear receptor activity in the introduction (P.3) to cover the point.

  1. The rationale showing in both tables only a very small selection of corepressors and coactivators is not clear. Why were these selected? Should the list be expanded?

Answer: Thank you for the comment. We understand that the list of transcription factor coregulators is vast. As per the available literature including one of our papers the list can go over 1400. Given the vast number of coregulators reported we are limited in their discussion in this review. Nevertheless, we selected coregulators based on literature available for nuclear receptor coregulators in different hormone dependent cancers and we believe that the ones we discussed represent some of the mostly studied in the relevant context.

  1. Why the nomenclature of are some nuclear hormone receptors is provided as NRs e.g. NR3C1 (GR) while others e.g., VDR? Please present homogenously names of nuclear receptors.

Answer: We apologize for the confusion. We have gone thoroughly over the manuscript and updated it to be consistent with all nuclear receptor abbreviations and nomenclature.

  1. Some discrepancies exist between text and tables. E.g. LCoR is mentioned in text to be a corepressor of AR but not mentioned in the table.

Answer: We have taken a thorough look into this and tried to include the missing information in both tables and text.

Reviewer 2 Report

This is a review manuscript about the nuclear receptor coregulators. The review provided relatively comprehensive knowledge about nuclear receptor coregulators and their impact on gene regulation. This review also covered biochemical and molecular techniques and tools that have been employed to identify nuclear receptor coregulatory proteins and analyze their functions. The review is well written.

However, a document uses several gene names that are not defined, making it hard to follow. Also, the author should provide brief information about Panther and FANTOM, which may not be familiar to many readers.

Author Response

We are thankful that reviewer highlight the biochemical and molecular techniques that we have covered in the review combined with the comprehensive knowledge of nuclear receptor coregulators. 

Document uses several gene names that are not defined, making it hard to follow. Also, the author should provide brief information about Panther and FANTOM, which may not be familiar to many readers.

Answer: Again, we apologize for limiting manuscript accessibility, and we have now described these resources in more detail and provided relevant references.

Reviewer 3 Report

Reviewer comments:

Comments to the Author

The authors have compiled the studies and importance of the mechanisms by which coregulators interact with and modulate the activity of nuclear receptors (NRs) providing opportunities to develop better prognostic and diagnostic techniques as well as therapeutic opportunities. This review aims to gather and summarize recent studies, techniques and bioinformatic methods used to identify distorted NR coregulator interactions that contribute as cancer drivers in hormone dependent cancers.

  • This review topic very relevant and is presented clearly and written in a well-ordered manner.

  • Authors have included the recent studies and supported their statements throughout the article.

Minor criticisms

  • Please provide a systematic figure for showing various functions governs by NRs to conclude section 1.3. Functions played by nuclear receptor coregulators.

  • Please describe more in details about Cointegrators.
  • Please undergo a thorough check of the manuscript for typographical and grammatical errors.

Author Response

Reviewer #3:

We appreciate the reviewer comment on the written and topics being clearly presented and well-ordered.

Minor criticisms:

  • Please provide a systematic figure for showing various functions governs by NRs to conclude section 1.3. Functions played by nuclear receptor coregulators.

Answer: A figure is now included on p.6

  • Please describe more in details about Cointegrators.

Answer: We have aimed to include more information pertaining to cointegrators

  • Please undergo a thorough check of the manuscript for typographical and grammatical errors.

Answer: We earnestly apologize that the manuscript contained typographical and grammatical errors, as well as omissions in the text; we understand how frustrating this can be for the reviewer. All authors have now thoroughly edited the manuscript and we believe that all these errors have been corrected.